# Trait-Specific Responses of Carabid Beetle Diversity and Composition in *Pinus densiflora* Forests Compared to Broad-Leaved Deciduous Forests in a Temperate Region

**Jong-Kook Jung [1],[*],[†]** and **Joon-Ho Lee [1],[2]**

[1] Entomology Program, Department of Agricultural Biotechnology, Seoul National University, Gwanakro 1, Gwanakgu, Seoul 08826, Korea; jh7lee@snu.ac.kr

[2] Research Institute of Agriculture and Life Sciences, Seoul National University, Gwanakro 1, Gwanakgu, Seoul 08826, Korea

[*] Correspondence: jk82811@korea.kr; Tel.: +82-2-961-2665; Fax: +82-2-961-2679

[†] Present address: Division of Forest Insect Pests and Diseases, National Institute of Forest Science, Seoul 02455, Korea.

**Abstract:** Since successful reforestation after the 1970s, Korean red pine (*Pinus densiflora*) forests have become the most important coniferous forests in Korea. However, the scarcity of evidence for biodiversity responses hinders understanding of the conservation value of Korean red pine forests. This study was conducted to explore the patterns of carabid beetle diversity and assemblage structures between broad-leaved deciduous forests and *P. densiflora* forests in the temperate region of central Korea. Carabid beetles were sampled by pitfall trapping from 2013 to 2014. A total of 66 species were identified from 9541 carabid beetles. Species richness in broad-leaved deciduous forests was significantly higher than that in pine forests. In addition, the species composition of carabid beetles in broad-leaved deciduous forests differed from that of *P. densiflora* forests. More endemic, brachypterous, forest specialists, and carnivorous species were distributed in broad-leaved deciduous forests than in *P. densiflora* forests. Consequently, carabid beetle assemblages in central Korea are distinctively divided by forest type based on ecological and biological traits (e.g., endemisim, habitat types, wing forms, and feeding guilds). However, possible variation of the response of beetle communities to the growth of *P. densiflora* forests needs to be considered for forest management based on biodiversity conservation in temperate regions, because conifer plantations in this study are still young, i.e., approximately 30–40-years old.

**Keywords:** biodiversity conservation; temperate forests; ground beetles; ecological trait

## 1. Introduction

The positive ecological role in plantation forests has been emphasized recently because it prevents the loss of biodiversity caused by deforestation worldwide [1,2]. For example, plantations can have direct impacts on biodiversity [3], as well as stand dynamics and structure [4]. The global plantation area is approximately 7.3% of the total forested area (291 million ha) [5]. However, conifer forests in Korea, mainly plantations, cover approximately 36.9% of the total forested area (2.3 million ha) [6]. Almost all natural forests in Korea had been destroyed until the 1960s and have recovered since the 1970s [7]. For this reason, a large area of forests in Korea is composed of 30–50-year-old conifer plantations and naturally regenerating deciduous forests covering approximately 87.3% of the total forested area [6]. Thus, understanding biotic responses of young forests, including plantations and

regenerated forests, provides valuable insight into biodiversity conservation in temperate forests of Korea.

Among young conifer plantation forests in Korea, *Pinus densiflora* forests are the most important, covering approximately 24.7% (1,562,843 ha) of the total forested area [6]. However, biodiversity in *P. densiflora* forests, as compared with young broad-leaved deciduous forests (*Quercus* spp.) in Korea, is poorly understood despite successful reforestation since the 1970s. In human-dominated landscapes, the diversity and composition of carabid beetles [8–11] and moths [12] in coniferous forests (mainly *P. densiflora*) have been investigated and compared to those in secondary mixed forests. In mountains in temperate regions, the effects of forest types on moths [13,14] and carabids [15–17] have been compared. Differences in carabid communities among forest types have been occasionally observed due to the effects of habitat fragmentation [8,11] and landscape heterogeneity [10]. However, the species diversity and composition of moths [13,14] and carabids [9,15–17] in coniferous plantations are generally similar to those in secondary or natural forests, except that the species richness of all carabid beetles is increased in regenerating deciduous forests [10]. In temperate regions, few studies have considered the ecological and biological traits of carabid beetles to compare diversity between forest types, such as habitat type [10,11], body size [10], and wing morphs [10,11,17]. By considering ecological and biological traits, the species richness of macropterous species differed only in grasslands as compared with various forest types, such as two natural forests (broad-leaved deciduous and *P. densiflora* forests) and one plantation (a deciduous coniferous forest) [17], and the species richness of brachypterous and forest specialists decreased significantly as a result of habitat fragmentation irrespective of forest type [11]. Nonetheless, the low number of spatial replications for measuring biodiversity in young *P. densiflora* forests is limited to emphasizing the conservation value compared to young broad-leaved deciduous forests because the distribution of insects is basically related to the local environment, such as habitat heterogeneity and elevation, in addition to habitat types.

To compare biodiversity in young *P. densiflora* and young broad-leaved deciduous forests, we studied carabid beetles because they are diverse, ecologically well known, and abundant in most ecosystems [18]. In particular, large-bodied and flightless carabid beetles are more vulnerable to disturbances than generalist species that have high mobility [19]. The endemism rate could also be an important biological characteristic because endemism is closely related to the low dispersal ability of carabid beetles combined with adaptation to specific local environments [20]. In addition, feeding guilds could also be an important trait, because diversity and distribution of carnivorous and herbivorous species are influenced by vegetation [21] and microhabitat characteristics, such as leaf litter [22]. These ecological and biological traits could provide a basis for biodiversity conservation in different forest types.

The primary aim of this study was to compare the abundance, species richness, and composition of carabid beetles based on ecological and biological traits (all carabid beetles, geographical distribution, wing morph, habitat preference, and feeding guilds) between young broad-leaved deciduous forests (*Quercus* spp. dominated, abbreviated to BLF hereafter) and young pine plantations (*P. densiflora* dominated, abbreviated to PF hereafter) in central Korea. In addition, we characterized carabid communities in each forest type by indicator species analysis, because carabid communities can not be regenerated even in relatively old plantations [23].

## 2. Materials and Methods

### 2.1. Study Area

Carabid beetles were studied in Yeongwol-gun and Jeongseon-gun, which are located in central Korea (Figure 1a). Because of the higher proportion of mountainous forests in the study area with steep terrain, forest landscapes are well preserved. In Yeongwol-gun and Jeongseon-gun, forests cover approximately 85% of the area [24]. The study area is surrounded by several reserves, such as Mt. Chiaksan National Park (17,567 ha), Mt. Sobaeksan National Park (32,205 ha), Mt. Taebaeksan

National Park (7005 ha), and forest genetic resource reserves on Mt. Gariwangsan (2475 ha) (Figure 1b). Moreover, the Donggang River traversing Yeongwol-gun, Jeongseon-gun, and Pyeongchang-gun has been designated an ecological and landscape conservation area (6497 ha) because of its beautiful landscape and high biodiversity. The climate of the region is temperate, with an average annual temperature of 11.03 °C; the average temperatures of the warmest and coolest months from 2011 to 2013 were 31.07 °C and −11.77 °C, respectively; the annual precipitation in 2011, 2012, and 2013 was 2085.7, 1398.8, and 1245.5 mm, respectively [25].

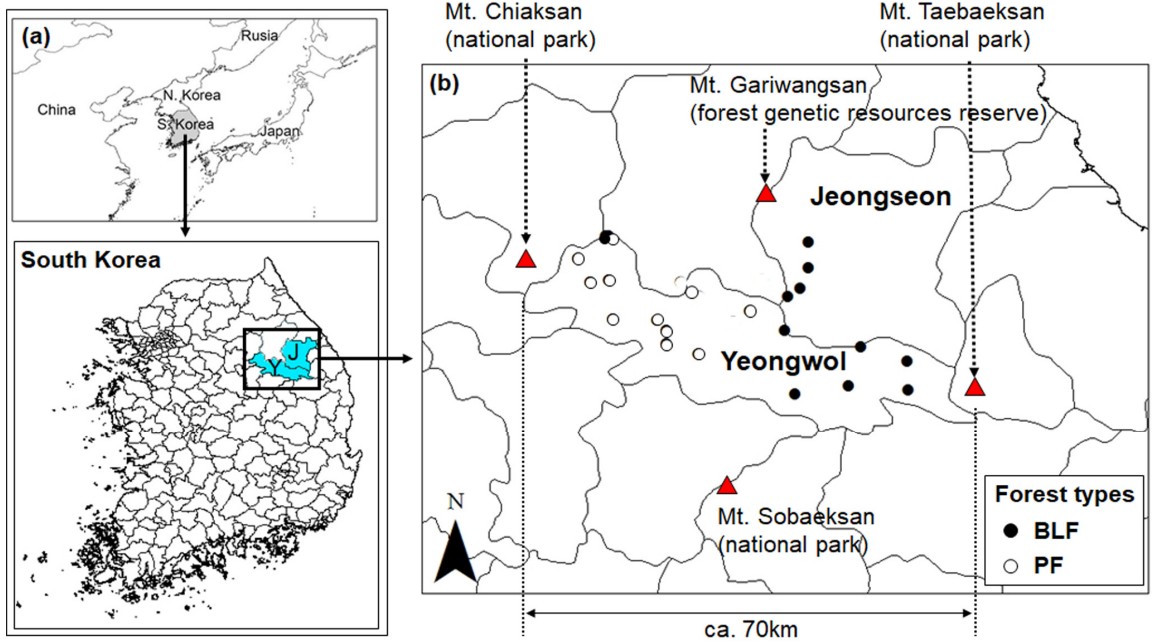

**Figure 1.** Map of (**a**) study area and (**b**) study sites (filled circle, broad-leaved deciduous forests (BLF); opened circle, *Pinus densiflora* forests (PF)).

Carabid beetles were sampled at 22 study sites (Figure 1b). The location and environmental characteristics of the study sites are described in Table 1. Half of the 22 sites were PF, the other half BLF. Forest information (forest types and ages) in this study area was confirmed by the Korean forest geographic information service system [6]. The latitude and longitude (i.e., spatial location) of the study sites were 37°05′03″ N, 37°22′23″ N and 128°13′48″ E, 128°48′35″ E, respectively, but the spatial locations of the two forest types were quite distinct (Figure 1b). In addition, forest ages of PF were relatively higher than those of BLF (Welch two sample t-test, $t = 3.49$, $p = 0.002$). However, elevation was not different between the two forest types (range of 245–473 m for PF and 273–744 m for BLF, $t = 1.49$, $p = 0.157$) (Table 1). These five variables for each sampling site were used in the redundancy analysis (RDA) to explain the variation in carabid beetle assemblages between forest types.

*2.2. Sampling*

Three pitfall traps were installed at each study site for collecting carabid beetles from 2013 to 2014. Sampling periods differed among study sites, ranging between 78–196 days (234–588 trap × days, Table 1). Short sampling periods at some study sites, such as YNC, YNG, and YBD, may have limited appropriate carabid beetle collection. However, we sampled carabid beetles at least from August to October at all sampling sites because the diversity of carabid beetles during these months in mountainous forests of central Korea was higher than that in other months [26].

**Table 1.** Environmental characteristics for each study site with sampling information.

| Location | | Code for Site | Forest Type [a] | Forest Age Class [b] | Latitude | Longitude | Elevation (m) | Trap × Days | Sampling Period |
|---|---|---|---|---|---|---|---|---|---|
| Jeongseon-eup | Gwangha-ri | DJGw | BLF | III | 37°21′39″ N | 128°37′43″ E | 296 | 330 | Jul–Oct 2013 |
| | Gasu-ri | DJGa | BLF | III | 37°18′49″ N | 128°37′41″ E | 285 | 330 | Jul–Oct 2013 |
| | Unchi-ri | DJU | BLF | III | 37°16′36″ N | 128°36′48″ E | 278 | 330 | Jul–Oct 2013 |
| | Deokcheon-ri | DJD | BLF | III | 37°15′43″ N | 128°35′23″ E | 273 | 330 | Jul–Oct 2013 |
| Yeongwol-eup | Geoun-ri | DYG | PF | V | 37°14′09″ N | 128°31′11″ E | 258 | 330 | Jul–Oct 2013 |
| Suju-meyon | Beopheung-ri | YSB1 | PF | IV | 37°22′18″ N | 128°15′45″ E | 473 | 330 | Jul–Oct 2013 |
| | Beopheung-ri | YSB2 | BLF | IV | 37°22′23″ N | 128°15′24″ E | 488 | 588 | Apr–Oct 2014 |
| | Unhak-ri | YSU | PF | V | 37°19′53″ N | 128°12′24″ E | 461 | 579 | Jul 2013–Jun 2014 |
| | Mureung-ri | YSM | PF | V | 37°17′28″ N | 128°15′50″ E | 280 | 579 | Jul 2013–Jun 2014 |
| Jucheon-myeon | Docheon-ri | YJD | PF | V | 37°17′16″ N | 128°13′48″ E | 323 | 579 | Jul 2013–Jun 2014 |
| | Yongseok-ri | YJYo | PF | IV | 37°13′12″ N | 128°16′14″ E | 355 | 579 | Jul 2013–Jun 2014 |
| Hanbando-myeon | Hutan-ri | YHH | PF | IV | 37°11′56″ N | 128°22′03″ E | 245 | 579 | Jul 2013–Jun 2014 |
| | Ongjeong-ri | YHO | PF | IV | 37°13′07″ N | 128°21′08″ E | 303 | 579 | Jul 2013–Jun 2014 |
| Jungdong-myeon | Jikdong-ri | YJJ | BLF | IV | 37°10′13″ N | 128°43′27″ E | 434 | 339 | Jun–Oct 2014 |
| | Yeonsang-ri | YJYe | BLF | IV | 37°12′01″ N | 128°35′07″ E | 332 | 339 | Jun–Oct 2014 |
| Sangdong-eup | Naedeok-ri | YSN | BLF | IV | 37°08′37″ N | 128°48′32″ E | 664 | 339 | Jun–Oct 2014 |
| | Deokgu-ri | YSD | BLF | IV | 37°05′29″ N | 128°48′35″ E | 744 | 339 | Jun–Oct 2014 |
| Gimsatgat-myeon | Nae-ri | YGN | BLF | IV | 37°05′53″ N | 128°42′03″ E | 525 | 339 | Jun–Oct 2014 |
| | Waseok-ri | YGW | BLF | III | 37°05′03″ N | 128°36′14″ E | 339 | 339 | Jun–Oct 2014 |
| Nam-myeon | Gwangcheon-ri | YNG | PF | V | 37°09′21″ N | 128°25′38″ E | 266 | 234 | Aug–Oct 2014 |
| | Changwon-ri | YNC | PF | V | 37°10′31″ N | 128°22′06″ E | 298 | 234 | Aug–Oct 2014 |
| Buk-myeon | Deoksang-ri | YBD | PF | III | 37°17′19″ N | 128°23′43″ E | 350 | 234 | Aug–Oct 2014 |

[a] Abbreviation of habitat types are: BLF, broad-leaved deciduous forests; PF, *Pinus densiflora* dominated forest; [b] Forest age classes in our study sties were confirmed by the forest geographic information service system in Korea (Korea Forest Service, 2016): III, 21–30-years old; IV, 31–40-years old; V, 41–50-years old.

The trap was a plastic cup (9.5 cm in diameter, 10 cm in height, 430 mL in volume), and a plastic roof was placed 3 cm above each trap to prevent the inflow of rainfall and litter. Pitfall traps were unbaited, containing preservatives (200 mL, 95% ethyl-alcohol/95% ethylene-glycol = 1:1) as killing-preserving solutions, which were replaced every 4 weeks. Collected carabid beetles were transported to the laboratory, dried, mounted, and identified to the species level under a dissecting microscope (63×, Olympus SZ61, Tokyo, Japan). Identification was performed according to [27–35]. Nomenclature confirmed the list of Korean Carabidae by [32–35]. Endemism with respect to Korea, including South and North Korea, was confirmed according to [32,34,36]. The identified carabid beetles were stored in the Insect Ecology Laboratory at Seoul National University.

### 2.3. Data Analysis

All of the following statistical approaches were performed in R version 3.3.2 [37]. The species richness of carabid beetles in each forest type was estimated by individual-based rarefaction curves using the "vegan" R package [38]. Rarefaction curves are based on a random resampling of the pool of captured individuals and are used to estimate expected richness at lower sample sizes [39]. Rarefaction methods enable meaningful standardization and comparison of datasets [39]. For the rarefaction curves, carabid beetle samples were pooled based on forest type and forest age.

To compare the abundance and species richness of total carabid beetles and functional groups between two forest types, a t-test was applied. For the t-test, carabid beetle samples were pooled at each site. In addition, we standardized species richness and abundance, dividing them by trap × days (ranging between 234 and 588 trap × days) for statistical analyses (Table 1) because sampling periods were different at each study site and some pitfall traps were disturbed by rain.

The RDA was performed to explain the variation in carabid beetle assemblages between forest types and to visualize differences in species composition between forest types. Because there is a possible "horseshoe effect" in ordination [40], a Hellinger transformation was applied to species data prior to RDA. To preselect the environmental variables, we applied the "ordistep" function from the "vegan" R package [41], which performs both forward and backward selection of variables based on *P*-values. The vegan function "anova" was used to assess the significance of each variable in the final model (sequential test) and to obtain the *P*-values for each variable based on permutation tests [42]. We further compared the species richness of different ecological groups of carabid beetles by performing a Pearson correlation analysis against plot scores of the first and second axes resulting from the ordination analysis (plot scores for RDA1 and RDA2 based on weighted averages of site scores).

The indicator value (IndVal) approach was conducted to find indicator species between forest types that could characterize the habitats [43]. Flexible IndVal was independent of the relative abundance of other species, and there was no need to use pseudospecies. IndVal is at maximum value (1.00) when all individuals of a species occur in a single group of sites and when the species is found in all sites of that group. Therefore, abundance and occurrence stability indices for species were determined for analysis. The statistical significance of the species indicator value ($\alpha$ = 0.1) was determined using the Monte Carlo permutation test [44,45].

## 3. Results

### 3.1. Diversity and Abundance of Carabid Beetles

A total of 66 species were identified from 9541 collected carabid beetles (Table S1). Three *Synuchus* species, i.e., *Synuchus nitidus* (3888 individuals, 40.75% of total), *Synuchus cycloderus* (2587 individuals, 27.11%), and *Synuchus agonus* (825 individuals, 8.65%), and *Eucarabus cartereti*, 1982 (418 individuals, 4.38%), were abundant, comprising over 80% of the total carabid beetle assemblages (Table S1). Fourteen endemic specieswere collected, comprising 8.84% of the total abundance (843 individuals).

Individual-based rarefaction curves indicated that the species richness of carabid beetles in BLF was higher than that in PF (Figure 2a). When considering forest age for each forest type, the species richness

of carabid beetles in 31–40-year-old BLF was more distinct than the other three forest types (Figure 2b). Considering ecological groups, the abundances of brachypterous, forest specialists, open-habitat species, and carnivorous species were not different between the two forest types, while those of widespread, macropterous, and herbivorous species were significantly higher in PF (Table 2). The abundance of endemic species was higher only in BLF. The species richness of endemic, brachypterous, forest specialist, and carnivorous species, in PF were significantly lower than those in BLF.

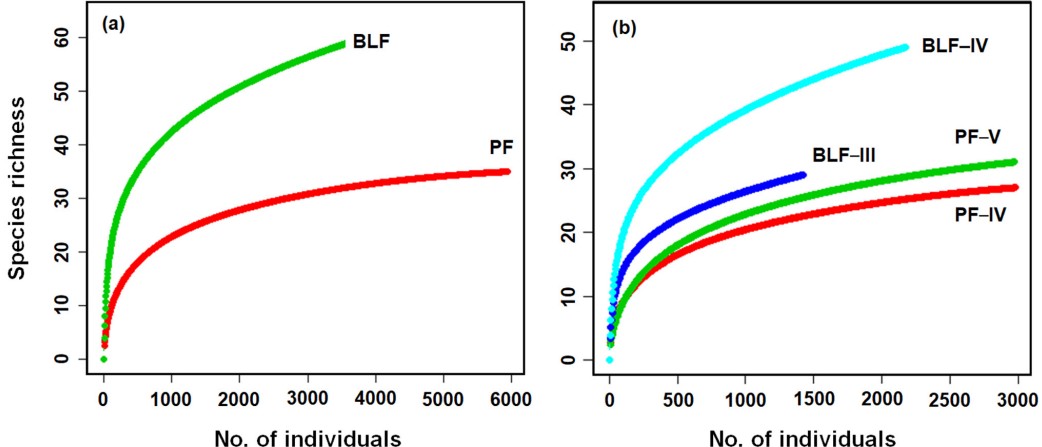

**Figure 2.** Individual-based rarefaction curves for ground-beetle catches (**a**) in *Pinus densiflora* forests (PF) and broad-leaved deciduous forests (BLF), and (**b**) in forest age classes (III, 21–30-years old; IV, 31–40-years old; V, 41–50-years old). Data points indicate average species numbers computed for the given number of individuals.

**Table 2.** Comparison of carabid catches (abundance and species richness) between *Pinus densiflora* forests (PF) and broad-leaved deciduous forests (BLF). Bold characters indicate statistical difference of abundance or species richness of carabid beetles between BLF and PF ($p < 0.05$).

| Dependent Variables | Carabid Catches (Mean ± S.E.) | | Statistics | |
|---|---|---|---|---|
| | PF (*n* = 11) | BLF (*n* = 11) | *t*-value | *p* |
| *Abundance* | | | | |
| All species | 540.6 ± 96.46 | 326.7 ± 77.63 | 1.728 | 0.1002 |
| **Endemic species** | **7.6 ± 2.47** | **69.0 ± 27.40** | **−2.231** | **0.0494** |
| **Widespread species** | **533.0 ± 95.44** | **257.7 ± 56.44** | **2.483** | **0.0243** |
| Brachypterous species | 53.3 ± 14.87 | 172.0 ± 70.11 | −1.657 | 0.1261 |
| **Macropterous species** | **487.4 ± 87.95** | **154.7 ± 33.31** | **3.537** | **0.0037** |
| Forest specialists | 519.0 ± 91.69 | 301.5 ± 79.25 | 1.795 | 0.0881 |
| Open-habitat species | 21.6 ± 12.47 | 25.3 ± 8.67 | −0.239 | 0.8135 |
| Carnivorous species | 539.5 ± 96.24 | 318.4 ± 75.34 | 0.942 | 0.3573 |
| **Herbivorous species** | **1.2 ± 0.38** | **8.4 ± 2.76** | **−3.002** | **0.0084** |
| *Species richness* | | | | |
| **All species** | **10.0 ± 0.88** | **15.4 ± 1.99** | **−2.466** | **0.0274** |
| **Endemic species** | **1.1 ± 0.31** | **4.5 ± 0.90** | **−3.534** | **0.0039** |
| Widespread species | 8.9 ± 0.72 | 10.9 ± 1.32 | −1.327 | 0.2038 |
| **Brachypterous species** | **4.5 ± 0.58** | **8.0 ± 1.41** | **−2.330** | **0.0362** |
| Macropterous species | 5.5 ± 0.78 | 7.4 ± 1.13 | −1.325 | 0.2021 |
| **Forest specialists** | **7.2 ± 0.60** | **11.3 ± 1.71** | **−2.262** | **0.0423** |
| Open-habitat species | 2.8 ± 0.72 | 4.1 ± 0.99 | −1.041 | 0.3116 |
| **Carnivorous species** | **9.2 ± 0.77** | **13.3 ± 1.75** | **−2.459** | **0.0265** |
| Herbivorous species | 0.82 ± 0.26 | 2.1 ± 0.55 | −0.9464 | 0.3629 |

### 3.2. Species Composition of Carabid Beetles

The RDA for carabid beetles was significant (permutation test for RDA under reduced model, $F_{4,17} = 3.51$, $p < 0.001$). The first and second axes accounted for 36.55% of the total variation, with the first and second axes explaining 24.17% (permutation test, $F_{1,17} = 7.51$, $p < 0.001$) and 12.37% ($F_{1,17} = 3.84$, $p = 0.005$), respectively (Table 3). The first axis of the RDA was strongly and positively associated with elevation and longitude, while forest type was negatively related to the first RDA axis (Figure 3 and Table 3). In addition, the species composition in BLF appeared to be distinct from that in PF (Figure 3). Species richness in most ecological groups showed positive correlations with weighted average site scores on the first RDA axis, whereas species richness of open-habitat species showed weak and positive correlations with the first axis (Table 3).

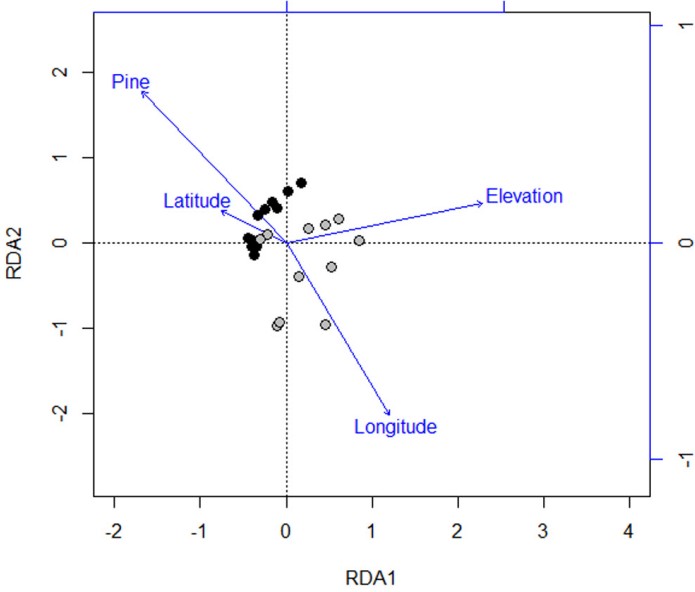

**Figure 3.** A RDA ordination of species composition of ground beetles in 22 sites. The first and second RDA axes are shown. The different forest types are shown as closed circles by different colors, i.e., black, broad-leaved deciduous forests and grey, *Pinus densiflora* forests.

**Table 3.** Correlations between environmental variables or species richness and the axes in the RDA analyses (total inertia = 0.3808). Species richness was calculated by biological or ecological groups.

| Variables | RDA 1 | RDA 2 | RDA 3 | RDA 4 |
|---|---|---|---|---|
| Eigenvalue | 0.0921 | 0.0471 | 0.0218 | 0.0114 |
| % variation explain [†] | 24.17 | 12.37 | 5.72 | 3.00 |
| Environmental variables | | | | |
| Forest type | ** −0.64 | *** 0.54 | −0.02 | −0.23 |
| Elevation | *** 0.87 | 0.14 | 0.05 | −0.32 |
| Latitude | −0.29 | 0.12 | *** 0.76 | 0.23 |
| Longitude | * 0.45 | ** −0.62 | −0.20 | −0.25 |
| Species richness | | | | |
| Total | *** 0.85 | −0.18 | 0.06 | −0.16 |
| Endemic | *** 0.86 | −0.07 | 0.08 | −0.14 |
| Widespread | *** 0.80 | −0.23 | 0.06 | −0.17 |
| Brachypterous | *** 0.79 | −0.06 | 0.09 | −0.20 |
| Macropterous | * 0.43 | −0.32 | −0.02 | −0.05 |
| Forest specialists | *** 0.83 | −0.01 | 0.02 | −0.18 |
| Open-habitat | 0.24 | * −0.43 | 0.08 | −0.04 |
| Carnivorous species | *** 0.76 | −0.11 | 0.04 | −0.20 |
| Herbivorous species | ** 0.55 | * −0.48 | 0.07 | 0.00 |

[†] Percentage = 100 × (variation explained by respective axis)/(variation explained by all environmental variables). Statistically significant correlations are indicated by * $p < 0.05$, ** $p < 0.01$, *** $p < 0.001$.

According to forest types, the following two characteristic groups of carabid beetle species were detected: (1) numerous in BLF, i.e., *Eucarabus cartereti cartereti*, *Harpalus discrepans*, *Pristosiavigil*, *Leistus niger*, and *Cymindis collaris*; (2) abundant in PF, i.e., *Coptolabrus smaragdinus branickii* (Table 4). However, *P. vigil*, *C. collaris*, and *C. s. branickii* had low indicator values ($p > 0.05$).

**Table 4.** Two-way indicator table showing carabid beetle species indicator value for habitat clustering hierarchy according to forest type, the number of individuals of ground beetle species by forest types, and their proportion in observation (%). Carabid species that showed high indicator value ($p < 0.1$) were listed.

| Forest Types | Indicator Value | *p*-Value | Number of Individuals (Proportion in Observation, %) | |
|---|---|---|---|---|
| | | | QF | PF |
| Broad-leaved deciduous forest (QF) | | | | |
| *Eucarabus cartereti cartereti* | 0.998 | 0.001 | 416 (100.0) | 2 (9.1) |
| *Harpalus discrepans* | 0.749 | 0.023 | 30 (63.6) | 4 (27.3) |
| *Pristosia vigil* | 0.729 | 0.060 | 77 (54.5) | 2 (18.2) |
| *Leistus niger niger* | 0.674 | 0.033 | 8 (45.5) | |
| *Cymindis collaris* | 0.603 | 0.093 | 8 (36.4) | |
| Pinus densiflora-dominated forest (PF) | | | | |
| *Coptolabrus smaragdinus branickii* | 0.699 | 0.081 | 6 (27.3) | 51 (54.5) |

## 4. Discussion

### 4.1. Carabid Beetle Diversity in Different Forest Types

In temperate forests of Korea composed of young coniferous and deciduous forests, we found that forest type and elevation were important factors that influenced carabid beetle assemblages. In particular, the species richness of carabid beetles of the BLF in this study was obviously higher than inPF, although the average elevation did not differ between the forest types. These results were rather different from those in previous studies conducted in Korea [15], China [16], and Japan [17], whereas similar findings were reported in Japan [46] and Korea [10]. In particular, the diversity of carabid beetles was significantly reduced in conifer plantations, especially in more heterogeneous landscapes [10]. In Japan, carabid beetles in fragmented landscapes at high elevations (1240–1490 m) showed no differences in diversity between natural forests (evergreen coniferous and broad-leaved deciduous) and plantations (deciduous coniferous) [17]. In contrast, similar carabid diversity between pine plantations and oak forests was reported from Korea and China [15,16], but the estimated species richness in mixed forests was higher than that reported in China [16]. On the basis of ecological traits of carabid beetles in several habitat types, including deciduous, mixed, and mature conifer forests, more forest specialists were caught in deciduous forests, while more forest generalists were caught in mature conifer plantations [46]. Thus, conifer plantations sometimes support biodiversity, but this is dependent on the history of disturbances such as forest management or biogeographical characteristics. In fact, many forest-inhabiting species in Korea are flightless, and thus they cannot disperse from source habitats to fragments, especially the large-bodied species [47]. However, our study region is composed of continuous mountainous forests, not fragmented landscapes, although the locations of the study sites of each forest type were not pairwise in the same locality. Thus, PF could have the potential to support low biodiversity, at least for carabid beetles.

Although this study showed that the diversity of carabid beetles clearly differed between forest types, the synergistic effect of elevation, locality of study sites, and forest type could also be important to understand carabid beetle assemblages in this region. Thus, interpretation of our results should be carefully applied to forest management for biodiversity conservation. This is because the difference in carabid communities could be due to the mixed effect of forest types and other environmental

variables, such as moisture, pH, organic matter, texture [17,48], leaf litter [22], canopy cover [16], and elevation [49].

### 4.2. Ecological and Biological Traits of Carabid Beetles and Forest Types

Plant species (i.e., forest type) are regarded as the primary factor for insect communities, especially herbivorous insects [50,51]. However, understanding the distributional pattern of carnivorous carabids in temperate forests could be more important as compared with phytophagous or omnivorous carabids because, in general, the biodiversity of carnivorous carabids is habitat heterogeneity dependent [52]. In fact, most carabid beetle species in this study were carnivorous (i.e., 9436 individuals belonging to 54 species), whereas only 105 belonging to 12 species were phytophagous or omnivorous (e.g., species belonging to genus *Amara* and *Harpalus*), which could occasionally occur in forests with understory vegetation because of forest gaps or forest roads. Thus, temperate forests in Korea could have no herbivorous carabid beetles. Nonetheless, forest type can alter other herbivorous insect communities, and carnivorous carabid beetles could change according to changes in the abundance and composition of herbivorous communities.

Unlike feeding guilds, this study suggested that the trait-specific response of carabid beetles to different forest types appears to be valuable information for establishing biodiversity conservation plans in young reforested landscapes. In this study, the species richness of endemic, brachypterous, and forest specialists in BLF was obviously higher than that in PF, while the abundance of macropterous and widespread species in PF was obviously higher than that in BLF. In general, highly heterogeneous topography due to mountains is regarded as one of the reasons for the high diversity of forest specialist carabid beetles in Korea. In fact, many forest specialists are brachypterous and they prefer stable environments, such as BLF in this study. PF with homogenized environments can be suitable habitats for macropterous and widespread species, such as *S. cycloderus* and *S. nitidus*. In fact, these two species, which are dominant in forests of South Korea [10], were significantly more common in PF than in BLF and accounted for approximately 68% of all carabid beetles in this study. Consequently, ecological and biological traits for carabid beetles are very useful to understand the distributional pattern of biodiversity in temperate forests and are beneficial for forestry through biodiversity conservation.

### 4.3. Habitat Specialists for Forest Types

This study showed that only three species (*E. cartereti*, *H. discrepans*, and *L. niger*) could be considered BLF specialists, although the species composition was quite different between forest types. However, some species or groups could have the potential to become bioindicators of distinct forest types. For example, species, belonging to the genus *Pterostichus* were more numerous in BLF (258 individuals belonging to 8 species) than in PF (69 individuals belonging to three species) (Table S1).

In contrast, there were no habitat specialists in PF, only *C. s. branickii* was more frequently observed in PF than in BLF. *C. s. branickii* is a habitat specialist in PF patches [11], and this species is also found in open habitats, such as agricultural fields, orchards, and lawns [53,54]. Thus, this species appears to be a habitat generalist. In the genus *Synuchus*, some species, such as *S. cycloderus* and *S. nitidus*, can also be used as potential bioindicators. Although they were found in almost every study site, these two species were abundant in PF. This could be largely due to the habitat preference of *Synuchus* towards dry forests [55]. In fact, *P. densiflora* trees are generally planted on south-facing slopes of mountains in Korea, and the trees prefer well-drained soils. In addition, *P. densiflora* is considered to be the most preferable tree species for plantations in Korea because of its esthetic value [56]. Thus, habitat conditions in PF are generally dry, especially in fragmented patches [11]. For these reasons, some *Synuchus* species, especially *S. cycloderus* and *S. nitidus*, are potential bioindicators in PF. Nonetheless, many PF in this study were still young, approximately 31–50-years old. Therefore, habitat specialists did not have enough time to establish populations in PF.

## 5. Conclusions

This study showed that carabid beetle assemblages in temperate forests of central Korea were distinctively divided by forest type based on ecological and biological traits (e.g., endemism, habitat types, and wing forms). This suggested that monoculture plantations in temperate regions (i.e., *P. densiflora* in this study), which appeared to be simple habitats, could have a limited ability to preserve high biodiversity, at least for carabid beetles. In particular, young *P. densiflora* plantations may not be appropriate for supporting populations of endemic, brachypterous, and forest specialist species. For biodiversity conservation in Korea as a reforested area, however, possible variation of the beetle community response to forest growth of *P. densiflora* plantations need to be considered.

**Supplementary Materials:** The following are available online at http://www.mdpi.com/1424-2818/12/7/275/s1, Table S1: List of carabid beetles with number of individuals in *Pinus densiflora* forests (PF) and broad-leaved deciduous forests (BLF) in central Korea.

**Author Contributions:** Conceptualization, J.-K.J. and J.-H.L.; Data curation, J.-K.J.; Formal analysis, J.-K.J.; Funding acquisition, J.-H.L.; Investigation, J.-K.J.; Methodology, J.-K.J.; Project administration, J.-H.L.; Visualization, J.-K.J.; Writing—original draft, J.-K.J.; Writing—review and editing, J.-H.L. All authors have read and agreed to the published version of the manuscript.

**Funding:** This research was funded by Brain Korea 21 plus.

**Conflicts of Interest:** The authors declare no conflict of interest.

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
