# Peer review of "Trait-Specific Responses of Carabid Beetle Diversity and Composition in Pinus densiflora Forests Compared to Broad-Leaved Deciduous Forests in a Temperate Region"

_diversity, doi:10.3390/d12070275_

Round 1

Reviewer 1 Report

This study documented teh carabid diversity in P.densiflroa plantations in mountaneous land in S Korea, anbd compared it to that of broad-leaved forests. The latter can supprot a godo part of the forest-specialist carabid fauna, including endemic species. The study is well executed and carefully evaluated.

I suggest that you rephrase and shorten the introduciton. Several editorial suggestions are on the attached pdf file.

Author Response

We revised as Reviewer's requested. Please see the attachment.

Reviewer 2 Report

The paper is a well written account of species differences and abundances in two types of forests. The authors do a nice job of describing each forest type or region and then describe the methods to test the beetle species that live in each area. The analyses are strong and the numbers of traps and locations is sufficient to draw the conclusions presented by the authors. The authors do lack some information about the biological and ecological benefits of the specific beetle species; this could be highlighted more to strengthen the argument about why to survey for these beetles and what a population of beetles may mean to a forest.

Table 1. Not sure of the style of the sampling period column. Is it year then month?

Line 171: Please refer to table A1 again after “and so on…” to let readers know where they can find the complete list of beetles.

Line 181: You may need to remind the reader of how you came up with the QF abbreviation. It just isn’t intuitive for a broad-leaved deciduous forest. Maybe BLF is better? Or you refer to it in the text as Quercus dominant instead of broad leaved?

Line 287: Either stick with the long form of your forest types or the PF and QF abbreviations after you’ve introduced them. Don’t switch back and forth.

Line 313: Make sure your supplemental table label is the same here and in your manuscript text.

Sections 4.2 and 4.3: Missing information about the beetles themselves besides that they are categorized as specialist or generalist. What biological factors about the beetles could drive them to be in these given areas (besides being flightless?). What about the benefits of having a given beetle in an area, but missing a beetle in another area? Is the argument that abundance means better conservation, better soils, better forests? If so, please make that argument clearer either in the introduction or discussion.

Author Response

(The authors gave the same response as above.)
